# Decoding efficacy and resistance space at a drug binding site

Simone Altmann [1], Cesar Mendoza-Martinez [1,2], Melanie Ridgway [1], Michele Tinti [1], Jagmohan S. Saini[1,2], Peter E. G. F. Ibrahim [1,2], Michael Thomas [1,2], Manu De Rycker [1,2], Michael J. Bodkin[1,2] & David Horn [1] ✉

Assessing and understanding the impacts of all possible mutations at a drug binding site remain challenging. Here we use multiplex oligo targeting for mutational profiling, and computational modelling, to decode efficacy and resistance space at the otherwise native binding site for an anti-trypanosomal proteasome inhibitor. We saturation-edit twenty codons in the *Trypanosoma brucei* proteasome and subject the resulting libraries to stepwise drug selection and codon variant scoring, yielding dose-response profiles for >100 resistance-conferring mutants. Codon variant scores are predictive of relative resistance observed using a bespoke set of mutants, while fitness profiling reveals otherwise extensive constraints on mutational fitness and resistance space. The resistance profile is predictive of routes to spontaneous drug resistance observed within 'accessible', single nucleotide mutational space, while in silico predictions are closely aligned with impacts on drug resistance observed in cellulo. Thus, multiplex oligo targeting facilitates assessment of all possible mutations at a drug binding site.

Interactions between drugs and their targets impact efficacy and, when altered by mutation, can result in resistance[1–3], undermining the durability of chemotherapeutics used against infectious diseases or in oncology. An improved understanding of the impacts of all possible mutations at drug binding sites would therefore facilitate drug discovery and resistance surveillance, but assessing these impacts remains challenging[4–8]. Insights into drug – target interactions typically emerge from a combination of empirical experimental observation in cellulo, structural studies, and computational modelling and prediction in silico. Currently, although powerful approaches for saturation mutagenesis, combined with multiplexed assays, are available, in cellulo data describing comprehensive sets of mutations at native drug binding sites are often lacking[9], and confidence in in silico predictions can suffer from unquantified limitations as a result.

Several CRISPR-based editing approaches have been used to assess drug resistance associated mutations in eukaryotic cells. However, these approaches employ Cas9 nucleases that can yield undesirable and imprecise recombinants, base editors that typically introduce <50% of all possible edits[6,10], or prime editors that currently display limited efficiency[4,5,7]. CRISPR-based approaches are also typically constrained by the availability of suitable protospacer-adjacent motifs in the editing window and necessitate the introduction of further changes at the target site to prevent multiple rounds of editing. These approaches also present multiplexing challenges in diploid cells since different edits will likely be introduced in each allele, complicating genotype to phenotype assessments; although this latter challenge can be mitigated by using haploid cells[11]. Deep mutational scans, widely used for multiplex analysis of variant effects[12], have also been used to assess drug resistance mechanisms. However, these approaches typically employ ectopic expression systems that may alter target protein abundance and/or expression patterns[13].

Alternative approaches for the delivery of codon edits to native genes are available. Indeed, editing using multiplex genome engineering[14], or oligo targeting[15], simply require the delivery of short

---

[1]Wellcome Centre for Anti-Infectives Research, Faculty of Life Sciences, University of Dundee, Dundee, UK. [2]Drug Discovery Unit, Faculty of Life Sciences, University of Dundee, Dundee, UK. ✉e-mail: d.horn@dundee.ac.uk

single-stranded oligonucleotides. These approaches are DNA cleavage-free, and CRISPR-independent, and can deliver the full range of possible base edits, albeit at relatively low frequency. These approaches have been described in *Saccharomyces cerevisiae*[14], in embryonic stem cells[16], and in the trypanosomatids, *Trypanosoma brucei, T. cruzi* and *Leishmania*[15]; parasitic protozoa that are transmitted by insects, causing neglected tropical diseases in humans, and a range of veterinary diseases. Indeed, we recently reported the development of oligo targeting for the assessment of drug resistance mechanisms in the trypanosomatids[15].

In this work, we scale the oligo targeting approach for saturation-editing of the codons encoding a drug binding site in the *T. brucei* proteasome β5 subunit. The proteasome is a promising drug target in the trypanosomatids[17–21], in malaria parasites[22], and against multiple myeloma[23,24]. Cryo-electron microscopy structures of the *Leishmania* proteasome reveal a common binding site for both GSK3494245[21] and LXE408[25], and anti-trypanosomal proteasome inhibitors are currently in clinical development against both visceral leishmaniasis, and *T. cruzi*, which causes Chagas disease. Multiplex oligo targeting, amplicon sequencing, and codon variant scoring, combined with computational modelling to predict impacts on ligand affinity and proteasome function, provide unprecedented insights into the relationships between a drug and a mutated, but otherwise native, drug target. The readouts reveal how each binding site residue, and all possible binding site mutations, impact drug efficacy and drug resistance.

## Results

### Mutagenesis at a drug binding site

The proteasome is a high-priority drug target and was selected for the development and testing of multiplexed oligo targeting in trypanosomes. The proteasome comprises four stacked heteroheptameric rings with the β subunits contributing six proteolytic active sites that reside in the central chamber. These sites are autocatalytically activated in the assembled structure via cleavage of N-terminal propeptides; at the glycine preceding the catalytic threonine in the *T. brucei* β5 subunit (Tb927.10.6080)[26]. Anti-trypanosomal proteasome inhibitors selectively target the chymotrypsin-like site in this subunit[17,21,25].

Oligo targeting in *T. brucei* involves delivery of single-stranded oligodeoxynucleotides (ssODNs) of ~50 b in length, with higher efficiency editing achieved when using 'reverse-strand' ssODNs[15]; a single allele is typically edited in *T. brucei*, which is diploid. To target the chymotrypsin-like site in the β5 subunit, we considered a pooled ssODNs delivery protocol but noted the potential for introduction of multiple edits in the same cell[14], which would allow for enrichment of bystander edits, complicating codon-based genotype to phenotype assessments. We, therefore, tested co-editing efficiency by delivering pairs of ssODNs to wild-type *T. brucei* cells, one designed to introduce a resistance-conferring edit (G98S - *GGA-AGC*), equivalent to G98S in *T. cruzi*[15], and a second designed to introduce a synonymous edit either ~100 (C63C, TG*C*-TG*T*) or 200 bp (V31V, GT*T*-GT*A*) upstream. Although an allele replacement frequency of only 0.003 % was reported previously in wild-type *T. brucei*[15], not readily detectable using Sanger sequencing, we readily detected synonymous edits in drug-resistant populations generated using this approach (Supplementary Fig. 1), indicating efficient co-editing at adjacent sites, likely at a common transcription or replication fork. Accordingly, and to avoid co-editing, we subsequently delivered each degenerate ssODN individually.

We selected compound DDD247 (compound 7a in ref. 21), a low nanomolar potency inhibitor, for detailed analysis, and used cryo-EM structural data[21] to identify amino acids located within 5 Å of the proteasome β5 subunit binding site (Fig. 1a). The cognate codons for these twenty amino acids were distributed over a region of ~500 bp in the *β5* gene. To assemble a DDD247 binding site mutant library, we designed twenty reverse-strand, 53-b, ssODNs, each with a centrally located

degenerate 'NNN' (N = A, C, T, G) codon (Fig. 1b; Source Data, sheet 1). Mismatch-repair, previously reported to supress editing efficiency ~50-fold[15], was transiently knocked down using *MSH2* RNA interference for 24 h and, to avoid co-editing at adjacent sites in individual cells (see above), each degenerate ssODN was delivered individually. We then pooled the resulting populations and split the pool to generate a pair of Multiplex Oligo Targeting (MOT) libraries. Given a calculated allele replacement frequency of >0.04% using this approach (Supplementary Fig. 2), we estimated an average yield of >1000 codon edits per ssODN. To assess editing at each of the targeted codons, we extracted genomic DNA before, and six hours after library assembly, PCR-amplified the edited region in the proteasome *β5* gene, and deep-sequenced the amplicons (Fig. 1b). A scan to quantify variant codons across the edited region revealed highly specific editing at all twenty targeted sites (Fig. 1c; Source Data, sheet 2). We concluded that all 1280 possible variants around the β5 subunit drug binding site were likely represented in our pooled MOT libraries. These results served to validate the edited MOT libraries and the codon variant scoring approach.

### More than 100 resistance-conferring base edits

To select drug-resistant mutants, the MOT libraries were grown with DDD247 at 8 nM; twice the $EC_{50}$ (Effective Concentration of drug to inhibit growth by 50%). The selected libraries were then split three ways and grown with DDD247 at either 40, 200 or 1000 nM (Fig. 2a). We extracted genomic DNA following selection at each drug concentration, PCR-amplified the edited region in the proteasome *β5* gene, deep-sequenced the products (Fig. 2a), and quantified variant codons. Principle Component Analysis of variant codon read counts revealed similar profiles for the replicate MOT libraries (Fig. 2b). An assessment of edits generating stop/nonsense codons ($n = 60$) or synonymous codons ($n = 61$), neither of which were expected to yield resistant cells, revealed negligible read counts across the dose-response profile (Fig. 2c). In contrast, a number of edits generating mis-sense (non-synonymous) codons yielded elevated read counts across the dose-response profile, at the Y113 codon for example (Fig. 2c).

The heatmap shown in Fig. 2d (upper panel) shows relative representation of all 1280 codon variants in the unselected library and across the dose-response profile (Source Data, sheet 3). We observed >100 resistance-conferring edits that were enriched following drug selection (Fig. 2d). By averaging read counts for sets of synonymous codons, we derived a second heatmap showing the behaviour of all 400 possible amino acid variants across the dose-response profile, revealing forty-six resistance-conferring edits that were enriched following drug selection (Fig. 2d, lower panel).

Those edits associated with drug resistance show that oligo targeting can be used to introduce all twelve possible base edits, with no apparent bias in terms of single-nucleotide edits (SNEs), double-nucleotide edits (DNEs), or triple-nucleotide edits (TNEs). Eight of the twenty sites targeted yielded edits associated with drug resistance, with G98, Y113, V128 and S132 emerging as resistance 'hotspots' (Fig. 2d). Conversion to asparagine yielded resistance in six cases and glutamine yielded resistance in four, while only conversion to proline or arginine failed to yield resistance.

We next selected a set of nine drug-resistant mutants to interrogate the predictive power of the MOT-library profile. We used the sequencing read count data to reconstruct virtual dose-response curves, and to derive virtual $EC_{50}$ values for these mutants (Fig. 3a; Source Data, sheet 4; see Methods). As can also be observed in Fig. 2d, codons for the same amino acid edit yielded highly consistent results. A bespoke set of ssODNs (Source Data, sheet 1) was used to introduce each selected codon-edit in otherwise wild-type cells, and a panel of mutants was assembled, a pair of independent clones for each edit. The mutants, comprising six SNEs (G98A, G*GA-GCA*; Y113H, *TAC-CAC*; D115N, *GAC-AAC*; D116E, GAC-GA*A*; V128M, G*TG-ATG*; and Y136F, T*AT-TTT*),

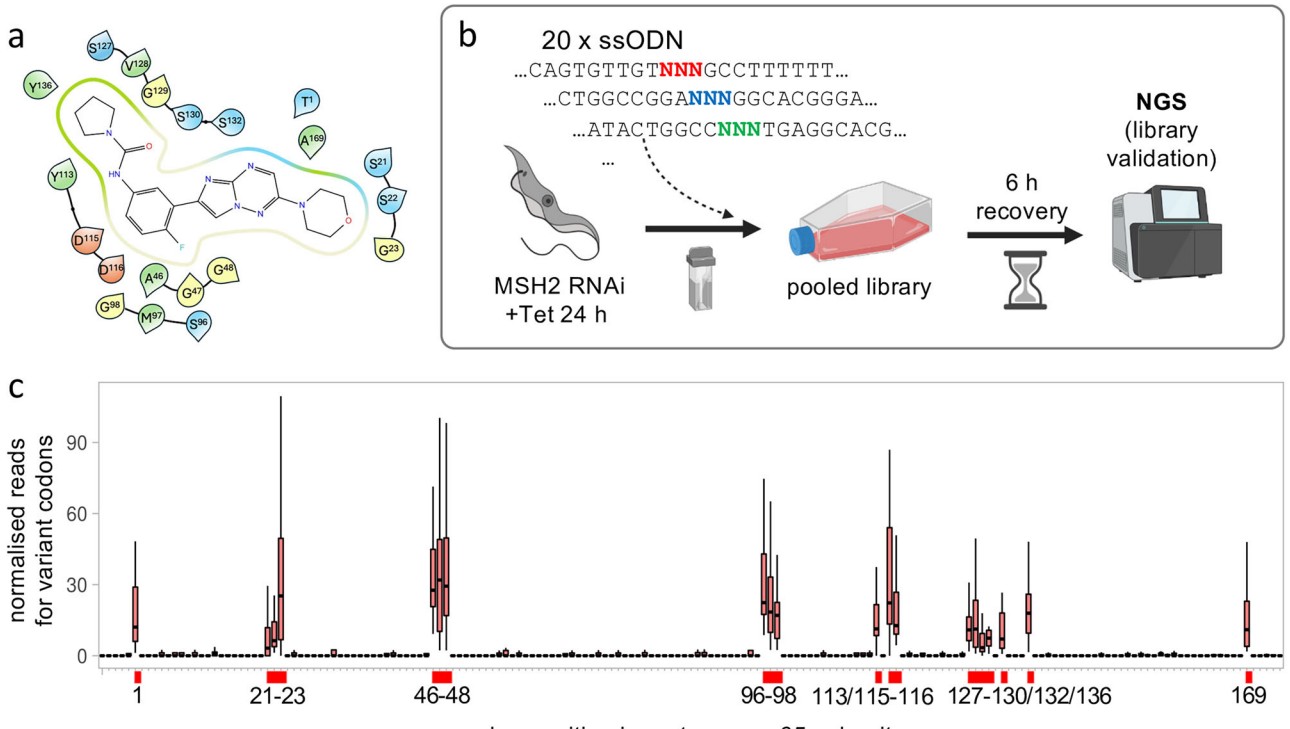

**Fig. 1 | Saturation mutagenesis at the proteasome β5 drug binding site. a** The ligand interaction diagram shows twenty *T. brucei* proteasome β5 subunit residues that are within 5 Å of the docked compound, DDD247. Green, hydrophobic; blue, positively charged; red, negatively charged; yellow, glycine. **b** The schematic illustrates the MOT-library assembly approach. Twenty single-stranded oligo-deoxynucleotides (ssODN) with a central degenerate codon were individually transfected into *T. brucei MSH2* RNAi cells 24 h after inducing knockdown with tetracycline. After 6 h, the pooled library and an untransfected control were assessed using deep sequencing of β5 amplicons. Created in BioRender. Altmann, S. (2025) https://BioRender.com/8sz8unx. **c** The boxplot shows specific editing for all twenty targeted codons, indicated on the x-axis, and as determined by deep sequencing and codon variant scoring; average of >33 M reads mapped per site for $n = 1$ pooled MOT library six hours after transfection relative to $n = 1$ population before transfection. Boxes indicate the interquartile range (IQR), the whiskers show the range of values within 1.5 × IQR, and a horizontal line indicates the median. Codon numbering is based on the mature β5 peptide.

a DNE ($V^{128}Q$ *GT*G-*CA*G) and two TNEs ($Y^{113}G$, *TAC-GGA*, $S^{132}M$ *TCC-ATG*), were validated by sequencing, confirming the expected edits in every case (Fig. 3b; Supplementary Fig. 3); these results also indicated that heterozygous edits were sufficient to yield resistance in every case. The mutant panel was then used to derive conventional dose-response curves and $EC_{50}$ values (Fig. 3c; Supplementary Fig. 4). A comparison with the virtual $EC_{50}$ data revealed an excellent correlation ($R^2 = 0.98$), indicating that codon variant scores from MOT libraries were indeed predictive of relative resistance observed using the bespoke set of mutants, and across a wide range of $EC_{50}$ values (Fig. 3d). Thus, we identified forty-six distinct drug-resistant mutants, which displayed up to 70-fold increased resistance relative to wild-type cells (Source Data, sheet 4), and which included all four previously known drug resistance mutations in trypanosomatid proteasome *β5* subunits; $Y^{113}F$ in *T. brucei*[25], $D^{115}N$ in *T. cruzi*[27], and $G^{98}C$ and $G^{98}S$ in *Leishmania*[21]. We concluded that MOT-library profiling can be used to identify large numbers of distinct drug resistance associated edits in trypanosomes, and also that sequence read counts for variant codons across a dose-response profile are excellent predictors of relative resistance.

As a further test of MOT-library profiling, we screened the libraries used above with bortezomib, which binds a site immediately adjacent to the DDD247 binding site (Supplementary Fig. 5a). Bortezomib is a modified Phe-Leu dipeptide with a boronic acid warhead[25]; the compound is approved for cancer chemotherapy, but resistance is a significant problem[23]. Selection of the MOT-library with 4 nM bortezomib, approximately twice the $EC_{50}$ for wild-type *T. brucei*, yielded resistant cells, while selection with higher drug concentrations failed to do so. Amplicon sequencing and codon variant scoring revealed only two conservative amino acid edits associated with bortezomib resistance,

$M^{97}I$ and $M^{97}V$ (Supplementary Fig. 5b; Source Data); edits that notably failed to yield DDD247 resistance. These results highlight the ability of MOT-library screens to discriminate between inhibitors that engage immediately adjacent sites within the same target.

### Fitness constraints on resistance space

More than 90% of edits fail to yield drug resistance in our screens, either because they fail to impact drug binding affinity, or simply because they yield defective proteasomes. Thus, mutational resistance space will be constrained by mutational fitness space at any given drug binding site. In order to assess mutational fitness space at the proteasome β5 subunit drug binding site, we deleted one copy of the *β5* gene, assembled MOT libraries in the diploid and haploid strains, and derived codon variant scores after six hours and four days, without any drug selection (Fig. 4a; Source Data). To validate the approach, we first assessed edits producing stop codons, which were severely depleted in the haploid strain, but could be retained in the diploid strain, as expected (Fig. 4b). This analysis also revealed dominant-negative impacts of more distal stop codons in the diploid strain, likely due to the expression of longer, albeit truncated, β5 peptides.

We next turned our attention to edits generating mis-sense (non-synonymous) codons, comparing relative codon-variant scores from the diploid and haploid strains, and for fifteen sites for which we obtained sufficient data. Mutational tolerance was reduced at all fifteen of these sites in the haploid strain relative to the diploid strain, indicating widespread loss-of-fitness associated with mutant proteasomes (Fig. 4c). Beyond this feature, three notable categories of edited codons emerged. First were codons for which most edits were poorly tolerated in both strains, consistent with dominant-negative defects

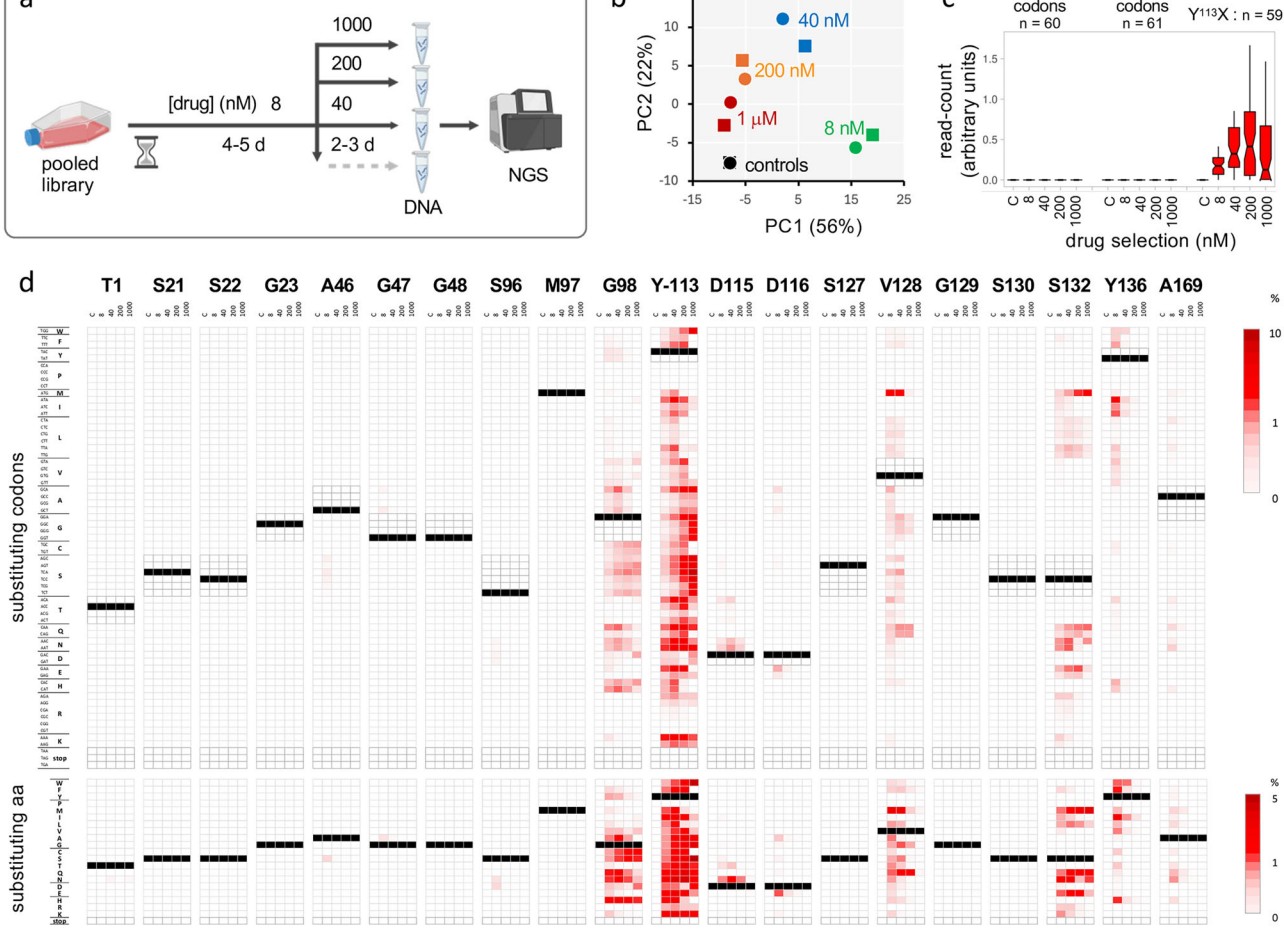

**Fig. 2 | Codon variant scoring reveals >100 resistance-conferring base edits.**
**a** The schematic illustrates the MOT-library profiling approach. The edited, pooled
*T. brucei* library described above was split in duplicates, which were subjected to
compound DDD247 selection as indicated. Genomic DNA was then isolated, and *β5*
gene amplicons were analysed by deep sequencing and codon variant scoring.
Created in BioRender. Altmann, S. (2025) https://BioRender.com/zy1spe1.
**b** Principal Component Analysis indicated that codon variant scores diverged from
the unselected control during DDD247-selection and that the replicate screens
yielded consistent results. **c** The boxplot shows codon variant scores for the indi-
cated categories and across the dose-response profile; averages from the two
replicate screens. Boxes indicate the interquartile range (IQR), the whiskers show

the range of values within 1.5 × IQR, and a horizontal line indicates the median. The
notches represent the 95% confidence interval for each median. **d** All sixty-four
possible codon variant scores for all twenty targeted sites in the proteasome *β5*
subunit gene are represented as a heatmap; for the unselected control sample, and
for averages from the two replicate screens across the dose-response profile (upper
panel); average of >6.5 M reads mapped per site. A heatmap for amino acid variant
scores is shown in the lower panel. Unedited codons are indicated (black), while
synonymous codons (upper panel only) and stop codons are highlighted with
darker framed outlines. Forty-six edits registered >0.1% of normalised reads at that
site following 8 nM selection.

($T^1$, $G^{48}$, $S^{127}$); second were codons for which many edits were tolerated
only in the diploid strain, consistent with loss of proteasome function
($A^{46}$, $G^{47}$, $M^{97}$, $D^{116}$, $V^{128}$, $S^{130}$, $Y^{136}$, $A^{169}$); and third were codons for which
many edits were tolerated in both strains, consistent with retained
proteasome function ($G^{23}$, $S^{96}$, $Y^{113}$, $S^{132}$) (Fig. 4c). Dominant-negative
edits were not expected to yield drug resistance, and indeed we saw no
evidence for resistance-associated edits at these three sites, which
included the catalytic threonine at $T^1$ (Fig. 2d). The second loss-of-
function category edits were also not expected to yield robust drug
resistance, and indeed resistance-associated edits were limited at these
eight sites, typically with failure to tolerate >40 nM drug selection
(Fig. 2d); $V^{128}$ did emerge as a resistance hotspot above but the mean
virtual $EC_{50}$ for edits at this site was only 23 nM ($n = 8$). In contrast,
edits in mutational fitness space, those associated with retained pro-
teasome function, have the potential to yield robust resistance, and
indeed we identified both $Y^{113}$ and $S^{132}$ as resistance hotspots above
(Fig. 2d). Thus, although only a minority of edits around the protea-
some *β5* subunit drug binding site sustain fitness, a substantial pro-
portion of these edits yield robust drug resistance. We concluded that

MOT-library profiling can be employed to elucidate both drug resis-
tance space at a given drug binding site, and constraints imposed on
resistance space by fitness space.

## Predicting resistance in SNP-accessible space
The codon-degenerate ssODNs used above were able to introduce the
full range of SNEs, DNEs, and TNEs, and to access all amino acid var-
iants. The vast majority of naturally occurring mutations are limited to
single nucleotides, however, and can access only nine alternative
codons, or five to seven alternative amino acids (see Fig. 5a, lower left
segment). Using the MOT-library profiling data to predict naturally
occurring mutations that confer compound DDD247 resistance, we
found that only fifteen of forty-six amino acid edits linked to drug
resistance above were accessible via an SNE (Fig. 5a, upper right seg-
ment, green squares); six are at the $Y^{113}$ resistance 'hotspot', while only
DNEs and TNEs were linked to resistance at the $S^{132}$ 'hotspot' (Fig. 5b).

Since generation of spontaneous proteasome inhibitor-resistant
trypanosomatid mutants typically takes several months[18,21,27], we took
advantage of transient knockdown of mismatch-repair[15] to rapidly

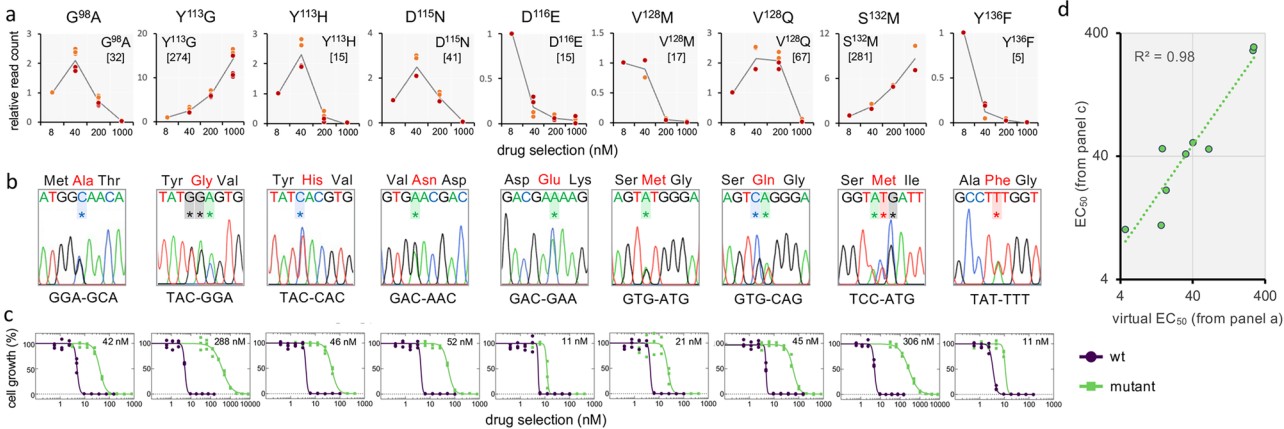

**Fig. 3 | Drug resistance assays reveal the predictive power of MOT-library profiling. a** The sequencing read counts underpinning codon variant scores were used to derive virtual dose-response curves for a selected set of edits; twenty codon edits representing nine distinct amino acid edits. We also derived a weighted average value from these data to yield virtual $EC_{50}$ values for compound DDD247 (nM), shown in square brackets. **b** A cognate panel of mutant clones was generated using oligo targeting in a wild-type background and all clones were validated by sequencing; data for one clone are shown here, and extended data for a second set of independent clones are shown in Supplementary Fig. 3. All edited bases are indicated with an asterisk. **c** Dose-response curves and $EC_{50}$ values for the panel of mutants. All compound DDD247 dose responses were measured in triplicate and representative dose-response curves for one biological replicate are shown here; data for the second set of biological replicates are shown in Supplementary Fig. 4. Average $EC_{50}$ values from both biological replicates are indicated. **d** The plot compares virtual $EC_{50}$ values from MOT-library profiling in (**a**) and average $EC_{50}$ values from the bespoke panel of mutants in (**c**), which are strongly correlated, $R^2 = 0.98$.

generate a panel of resistant *T. brucei* mutants. We sub-cloned the inducible mismatch-repair knockdown strain, induced knockdown for 24 h, subjected each clone to either 40 nM or 100 nM drug pressure for 6-9 days, sub-cloned each resistant population, and sequenced the *β5* gene in a single clone from each population. Sequence analysis revealed that twenty of twenty-three of these independent clones had mutations in the *β5* gene, all of which were single nucleotide mutations (Fig. 5c), as expected. Thirteen were V[128]M (G*TG*-ATG) mutants, which may have arisen frequently due to the presence of a native editing template; we identified a potential A-mismatched template sequence (TGAAATTTTTAGT*A*T) on four distinct chromosomes. Notably, this mutant also registered a particularly strong resistance signal in Fig. 2d. Although there are twice as many possible transversions (A or G - C or T) as transitions (A - G or C - T; see Fig. 5a, b), transitions, like the V[128]M mutation, are more frequent. Indeed, assessment of 355,892 polymorphisms[28] suggested that transitions are 1.8 times more frequent than transversion in *T. brucei*. Accordingly, the other seven *β5* subunit mutants comprised six transitions (Y[113]C, T*AC*-T*GC*; Y[113]H, T*AC*-*CAC*; D[115]N, G*AC*-*AAC* \*4) and a transversion (V[128]G, G*TG*-*GGG*) (Fig. 5c, Supplementary Fig. 6a), while the remaining three resistant clones had single nucleotide mutations in the proteasome *β4* subunit (Tb927.10.4710), all at F[24]L (Supplementary Fig. 6b), which also interacts with compound DDD247[21].

The resistance-associated mutation profile observed in *T. brucei* using MOT-library profiling (Fig. 2d) was also predictive of spontaneous drug resistance mutations reported previously in trypanosomatid proteasome *β5* subunits; Y[113]F in *T. brucei*, which conferred resistance to LXE408[25], D[115]N in *T. cruzi*, which conferred cross-resistance to DDD248, GNF6702, and TCMDC-143194[27], and G[98]C and G[98]S in *Leishmania*, which conferred resistance to DDD248[21]. All of these mutations are accessible via single nucleotide changes; with G[98]C (GGC-*TGC*) and G[98]S (GGC-*AGC*) only accessible via single nucleotide mutations in *Leishmania* due to distinct codon-usage at this site, which is GGA in *T. brucei*. In summary, all spontaneous DDD247-resistant *T. brucei* clones had a single nucleotide mutation in a proteasome subunit, with 87 % in the *β5* subunit; all these *β5* subunit mutations were predicted by MOT-library profiling (Figs. 2d and 5a); and no β5 subunit mutations were identified in residues more than 5 Å from the drug binding site (Fig. 1a). These observations, taken together, indicate that

MOT-library profiling in *T. brucei* can facilitate accurate prospective prediction of spontaneous drug resistance mutations in multiple trypanosomatids, and for multiple anti-trypanosomal proteasome inhibitors.

## Modelling ligand affinity

For computational modelling of ligand - proteasome binding affinity, we built a homology model of the *T. brucei* proteasome (Fig. 6a), revealing a high degree of similarity between the ligand binding sites in *Leishmania*[21] and *T. brucei* (Fig. 6b). Docking calculations focused on the cavity formed by residues within 5 Å of the DDD247 ligand (Supplementary Fig. 7a, b) and yielded excellent correspondence with the published *Leishmania* - GSK3494245 structure (RMSD average: -0.35 Å)[21], while docking of compound DDD247 yielded a similar pose, with very good alignment to the aromatic and aliphatic rings (RMSD average: -0.35 Å) (Supplementary Fig. 7c). Molecular dynamics simulations of the complex with compound DDD247 revealed stabilisation of ligand-protein interactions by hydrogen bonding with β5 residues, Y[113], G[129], and S[130], in particular, and with other hydrophobic interactions also present along the simulation, with β5-Y[113] and β4-F[24]; while the morpholine group displays evidence of being rather flexible and solvent exposed (Supplementary Fig. 8a, b).

We next considered all possible mutations in those twenty β5 subunit residues within 5 Å of the DDD247 ligand. After clustering snapshots from the molecular dynamics simulation trajectory (Supplementary Fig. 9), selected clusters were used for ΔΔG Molecular Mechanics General Born Surface Area (MM/GBSA) calculations[29], revealing G[98], Y[113], V[128], and S[132] as hotspots where mutations were predicted to decrease ligand affinity (Fig. 6c). Remarkably, these were the same four hotspots identified by in cellulo MOTlibrary profiling above (Fig. 2d). Among these hotspots, we observe ligand displacement at G[98] and S[132] (Fig. 6d). Indeed, the resistance-associated mutations we observe in cellulo all increase the side-chain mass at these sites; by 58 +/− 31 Da at G[98] ($n = 7$) and by 35 +/− 8 Da at S[132] ($n = 7$). Exemplar images illustrate how G[98] and Y[136] mutations have substantial or minimal impacts on ligand binding, respectively (Fig. 6e).

Hydrogen-bonding and lipophilicity are also predicted to have substantial impacts of protein-ligand affinity (Fig. 6f). Both features are predicted to contribute to the impact of Y[113] mutations, since this

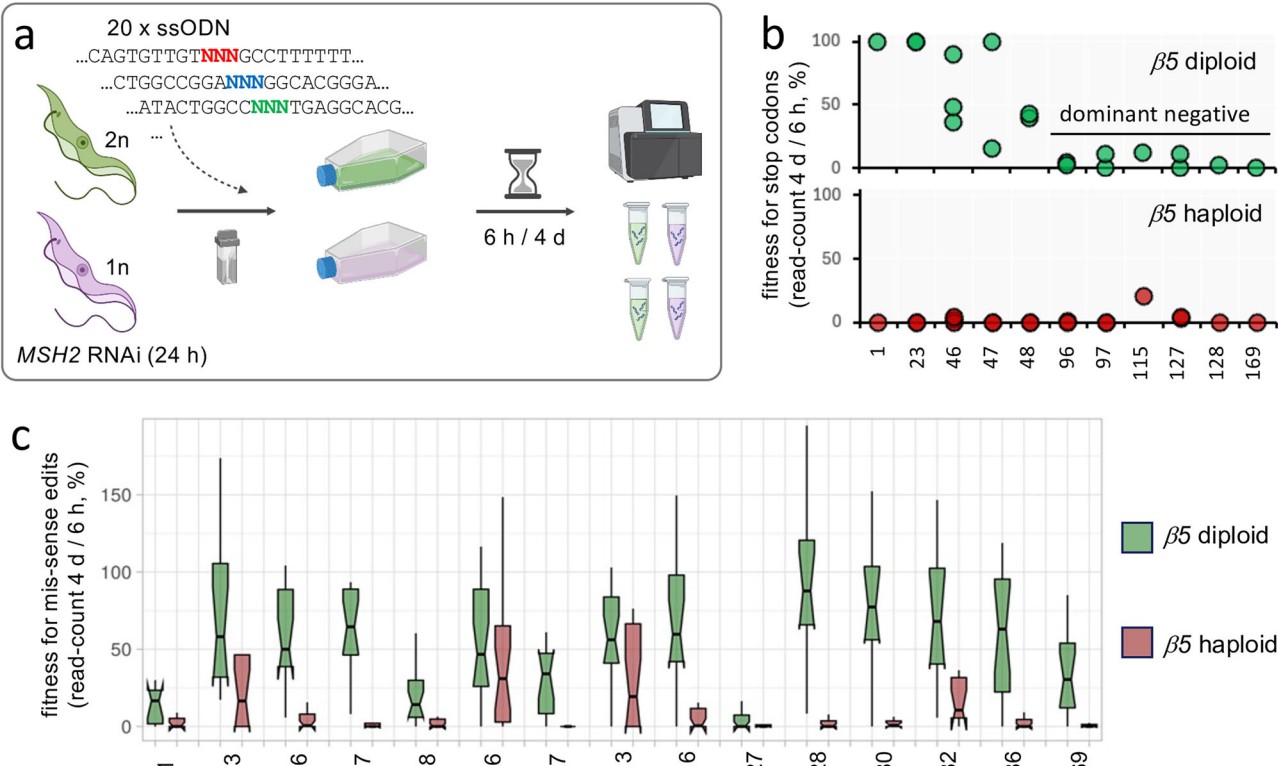

**Fig. 4 | Constraints imposed on mutational resistance space by fitness space.**
**a** The schematic illustrates the MOT-library fitness profiling approach. The set of twenty ssODNs with degenerate codons were individually transfected into *T. brucei MSH2* RNAi cells, either with two (diploid, 2n, green) or one (haploid, 1n, purple) proteasome *β5* allele, 24 h after inducing knockdown with tetracycline. Cultures were assessed using deep sequencing of *β5* amplicons and codon variant scoring prior to transfection and in duplicate, 6 h after transfection and 4 d after transfection, for each genotype. Created in BioRender. Altmann, S. (2025) https://BioRender.com/wi23cs9. **b** The plots show fitness, as revealed by the relative percentage of reads for stop codon edits remaining 4d after ssODN transfection, and at the sites indicated in the proteasome β5 subunit; in the diploid strain (upper panel) or in the haploid strain (lower panel). **c** The boxplot shows fitness, as revealed by the relative percentage of reads for mis-sense (non-synonymous) edited codons remaining 4d after ssODN transfection, at the sites indicated in the proteasome β5 subunit; averages from two replicate screens and >27 M reads mapped per site. Boxes indicate the interquartile range (IQR), the whiskers show the range of values within 1.5 × IQR, and a horizontal line indicates the median. The notches represent the 95% confidence interval for each median.

residue can form a hydrogen bond with the ligand and also interact within a lipophilic environment. Mutations at G[98], although they do not form a hydrogen bond with the ligand, all disrupt hydrogen bonding between the adjacent Y[113] residue and the ligand, while impacts on lipophilicity appear to be the major drivers of drug resistance for V[128] mutations. In the case of S[132], molecular dynamics clusters often placed this residue distal from the DDD247 ligand binding pose. Cluster 8, however, indicated a direct hydrogen bond between S[132] and the ligand, and this bond is lost in mutants associated with drug resistance. Accordingly, we suggest that this hydrogen bond does occur but is not observed in the cryo-EM structure. By comparing these ΔΔG (MM/GBSA) calculations with the full set of forty-six virtual EC$_{50}$ values derived using MOTlibrary profiling above, we obtained an $R^2$ value of 0.42 (Fig. 6g). As an example of excellent correlation between observation in cellulo and prediction in silico, Y[113] mutations incorporating highly charged residues (Y[113]K, Y[113]D, Y[113]Q, Y[113]S) are associated with a higher degree of resistance relative to mutations that conserve hydrophobicity (Y[113]F, Y[113]L, Y[113]H, Y[113]M).

## Modelling drug resistance

We elucidate above the mutational fitness space that restricts mutational resistance space at the DDD247 binding site; whereby non-functional mutant proteasomes fail to yield drug-resistant cells (Fig. 4). We, therefore, sought to combine in silico predictions of loss of protein function with our ligand affinity predictions. We used Evo 2, a DNA language model trained on >9 trillion DNA base pairs

from all domains of life[30], to predict loss of proteasome function due to codon changes, considering the full set of 1280 mutations surveyed by MOT-library profiling above. Evo2 Δ_scores (Supplementary Fig. 10) were used to assign penalties relative to loss-of-function for each mutated β5 subunit residue. Both the affinity scores (Fig. 7a, left-hand panel) and functional scores (Fig. 7a, right-hand panel) were then combined to generate a heatmap showing resistance predictions (Fig. 7a, lower panel); for those mutations that both disrupt ligand affinity and retain fitness. The resistance hotspots described above, G[98], Y[113], V[128] and S[132], were retained, and the computationally predicted impact of each individual mutation can now be seen in the combined heatmap. Although the analysis predicted reduced ligand affinity due to mutations at A[46] and G[47], loss of proteasome function is also predicted due to these mutations, meaning that resistance is predicted to be minimal (Fig. 7a). Indeed, mutations at these sites registered a substantial loss-of-fitness in cellulo (Fig. 4c) and failed to register as resistance mutations (Fig. 2d) in our in cellulo screen. Further highlighting the potential for added value when combining affinity and function parameters to predict drug resistance, Pearson's $r$ for the full dataset was increased from 0.46 to 0.51 after applying penalties for loss-of-fitness; Pearson's $r$ for loss-of-fitness alone was 0.01. A circular plot shows the full in cellulo resistance profile and in silico predicted resistance profile side-by-side for comparison, and a correlation plot shows how effectively our in silico analyses predicted the in cellulo derived resistance hotspots (Fig. 7b, Pearson's $r = 0.9$).

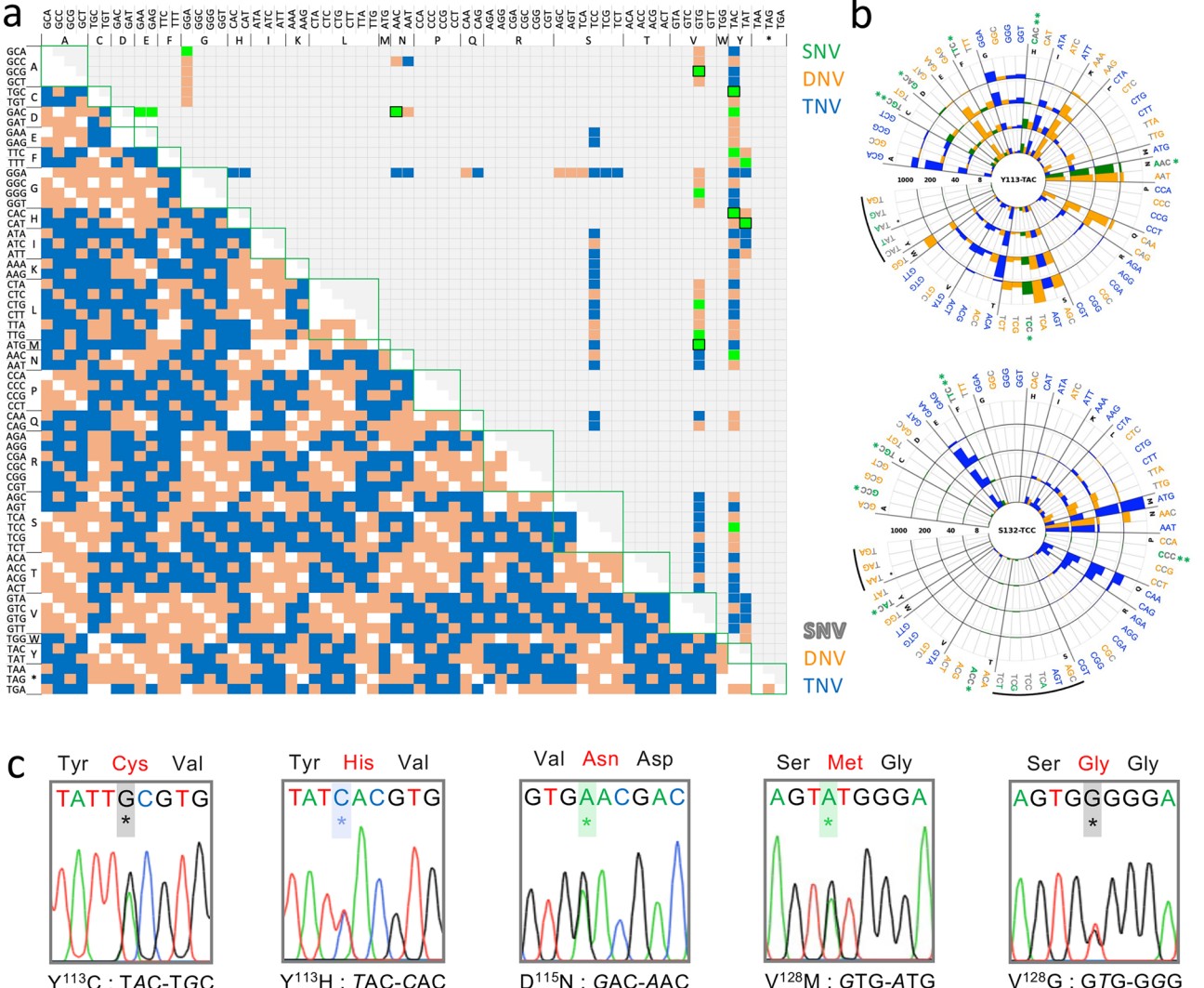

**Fig. 5 | Accurate prediction of drug resistance mutations in SNP-accessible space. a** The lower left-hand half of the plot illustrates the mutability of a given codon into any other codon via exchange of one (single nucleotide variant, SNV, white), two (double nucleotide variant, DNV, orange) or three (triple nucleotide variant, TNV, blue) nucleotides. The upper right-hand half of the plot shows those codon edits observed in our DDD247-resistance MOT-library screening profile, with SNVs shown in green, and transitional SNVs also framed. **b** The radial plots show dose-response data for two DDD247-resistance hotspots

observed in the proteasome β5 subunit in our MOT-library profiling screens (Fig. 2d), indicating that resistance is only accessible via SNVs at the Y[113] hotspot. TAC[113Y] can access TGC[C], GAC[D], TTC[F], CAC[H], AAC[N] and TCC[S] via an SNV, all predicted to yield resistance. TCC[132S] can access GCC[A], TGC[C], TTC[F], CCC[P], ACC[T] and TAC[Y] via SNVs, but none of these are predicted to yield resistance. Transitions, **; transversions, *, black arcs denote synonymous and stop codons. **c** Sanger sequencing data for five spontaneous and predicted SNVs in the proteasome β5 subunit associated with DDD247-resistance. *, indicates mutated bases.

## Discussion

Key goals in drug discovery are to evolve potent, safe, and durable therapies. Optimisation of drug efficacy typically requires substantial investment, however, with medicinal chemistry often informed by structural data and modelling when the target is known. Despite the importance of understanding drug – target interactions, and their impacts on efficacy and resistance, the relevant interactions often remain hypothetical, speculative, and incompletely characterised; in particular, empirical in cellulo data regarding individual residues at a drug binding site are often unavailable.

To address this knowledge gap, we sought to scale the oligo targeting approach in *T. brucei*[15]. We focused on the proteasome, an established drug target in diverse cell types[18,21–24], and developed Multiplex Oligo Targeting for site saturation mutagenesis. Guided by cryo-electron microscopy structures of the ligand-bound proteasome[21], we identified residues within 5 Å of the ligand and constructed libraries of cells with all possible mutations at these sites.

Following drug selection, we profiled all drug resistance edits using deep sequencing and variant scoring. The genome editing approach used here is rapid and precise, allows for expression of mutants from the otherwise unperturbed native locus, and offers high fidelity multiplexed assessment of edits quantified directly at the target locus. We do not observe homozygous edits using oligo targeting here, and this may be a limitation for investigating some drug resistance mechanisms, but we note that the heterozygous edits we observe most likely reflect those potential routes to drug resistance in a clinical setting. Thus, the approach facilitates decoding of efficacy and resistance space at a given drug binding site.

The current *T. brucei* libraries incorporated 1280 alternative codons encoding 400 proteasome β5 subunit variants. In this case, our goal was to introduce every possible alternative codon at each of twenty targeted sites. Based on an allele replacement frequency of ~0.04% following transient suppression of mismatch repair, we estimated 26,000 edited cells among 65 million cells in a 150 ml culture

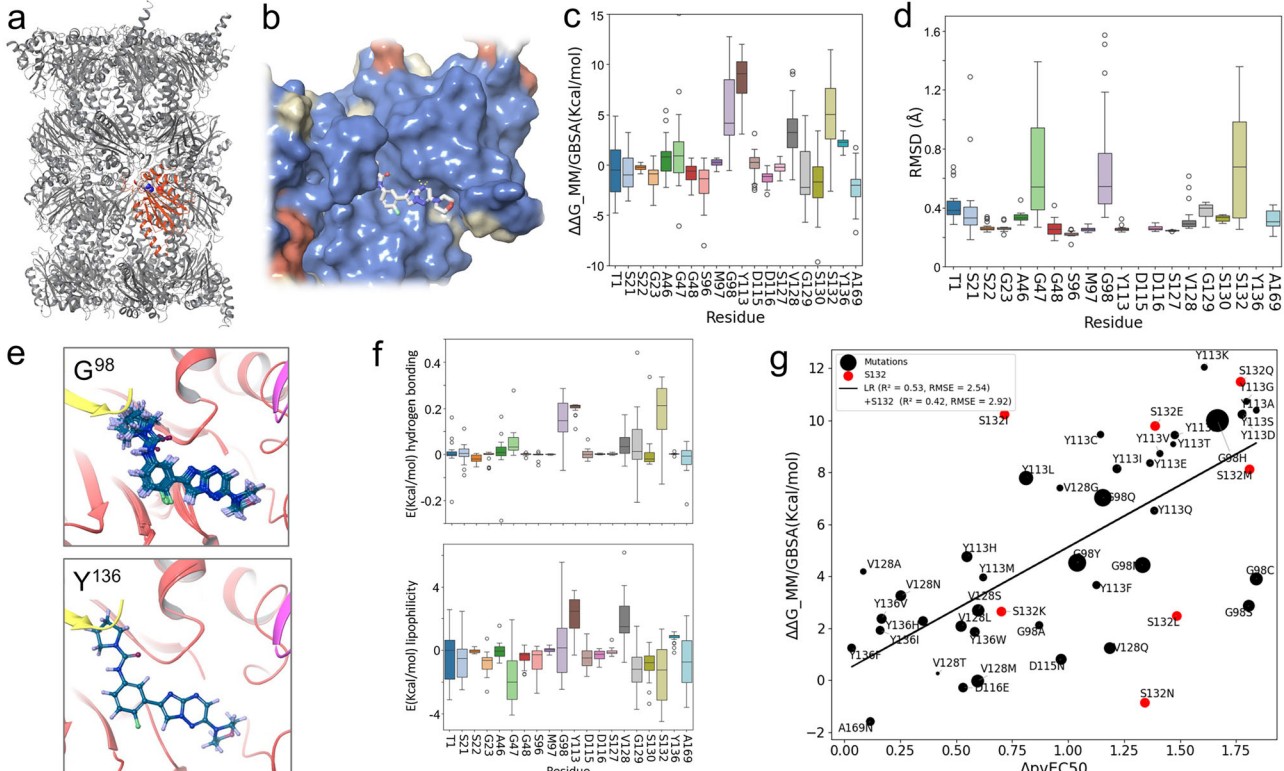

**Fig. 6 | Computational modelling of ligand affinity for mutant proteasomes.**
**a** The *T. brucei* proteasome homology model, highlighting the β5 subunit in red and the DDD247 ligand binding site. **b** The ligand DDD247 binding site is surrounded by residues that are conserved between *Leishmania* and *T. brucei*. Conserved, blue; similar, yellow; dissimilar, orange. **c** The boxplot shows the impact of mutations, $n = 20$ alternative amino acids, on ligand affinity at the sites indicated, as predicted using MM/GSA free energy calculations. Boxes indicate the interquartile range (IQR), the whiskers show the range of values within $1.5 \times$ IQR, and a horizontal line indicates the mean. **d** The boxplot shows RMSD for the ligand predicted for mutations, $n = 20$ alternative amino acids, at each site and relative to the native conformation. Other details as in (**c**). **e** The examples show the impact of mutations at $G^{98}$ and $Y^{136}$, where the ligand is overlaid from the final minimisation of all possible mutations. **f** The boxplots show the calculated binding free energy from c decomposed into hydrogen bond and lipophilicity contributions. Other details as in (**c**). **g** Correlation between binding affinity ($\Delta G\_MM/GBSA$) and virtual $EC_{50}$ values calculated from MOT-library profiling data ($\Delta pvEC_{50}$). $R^2$ is 0.42 for all sixty-four mutations analysed, and 0.53 when $S^{132}$ mutations are excluded (red). Data-point size indicates SD for clusters in set 1, while data for $S^{132}$ mutations were derived from cluster 8 (Supplementary Fig. 9).

volume, which equates to an average twenty-fold redundancy for each alternative codon. For comparison, a library generated recently using prime editors in human cells incorporated 2476 single nucleotide edits encoding 1726 epidermal growth factor receptor variants[7]. In terms of scaling, our results suggest that larger libraries could readily be generated and screened using the oligo targeting approach by simply using larger numbers of degenerate ssODNs. Substantial further scaling would present challenges in terms of assessing every alternative codon at each targeted site, but use of larger cell numbers and culture volumes would reduce bottlenecks and maintain library complexity.

Since drug resistance space is constrained by fitness space, we also assessed the fitness of cells exclusively expressing edited proteasomes. The majority of proteasome edits came with a fitness cost, and indeed, these edits failed to yield robust resistance. Since spontaneous drug resistance space is further constrained by accessible mutational space, a better understanding of these parameters will facilitate prediction and surveillance, as well as the prospective design of second-line therapies that bypass anticipated resistance mechanisms. Ultimately, and highlighting the potential value of such insights, compounds that target sites within highly restricted fitness space and otherwise inaccessible resistance space should deliver more durable therapies. Compounds identified as particularly vulnerable to spontaneous resistance could also be prioritised for use in combination therapies[31].

Computational models and artificial intelligence-based approaches promise to revolutionise the design of more active and durable drugs. In silico predictions often remain unvalidated due to the limited availability of in cellulo experimental data, however. By combining empirical data with in silico predictions of drug target affinity and proteasome structure - function perturbing mutations, we demonstrated how effective these in silico approaches can be. Notwithstanding some limitations, the computational models we present yield relatively accurate predictions of drug target affinity and resistance, explained by a combination of steric effects, hydrogen-bonding and lipophilicity. Such models will require comprehensive, high-quality training data to further improve their utility in future drug design.

Resistance could undermine the efficacy of new anti-parasitic drugs currently in pre-clinical or clinical development[32,33], and bridging the knowledge gap in relation to mutations that impact drug – target interactions is consequently a priority. Although rapid advances in structural biology and computational drug design present new opportunities in this area, the empirical in cellulo data describing comprehensive sets of mutations at native drug binding sites are often lacking. We developed multiplex oligo targeting in *T. brucei* and used the approach to saturation edit a native drug binding site in the proteasome β5 subunit. We profiled both fitness and resistance space and identified resistance hotspots and >100 resistance-conferring mutants, which allowed us to accurately predict spontaneous resistance mutations in multiple trypanosomatids. Assessment of the empirical data using computational modelling provided unprecedented insights into anti-trypanosomal proteasome inhibitor – target interactions. By further assessing the impacts of all possible mutations at drug binding

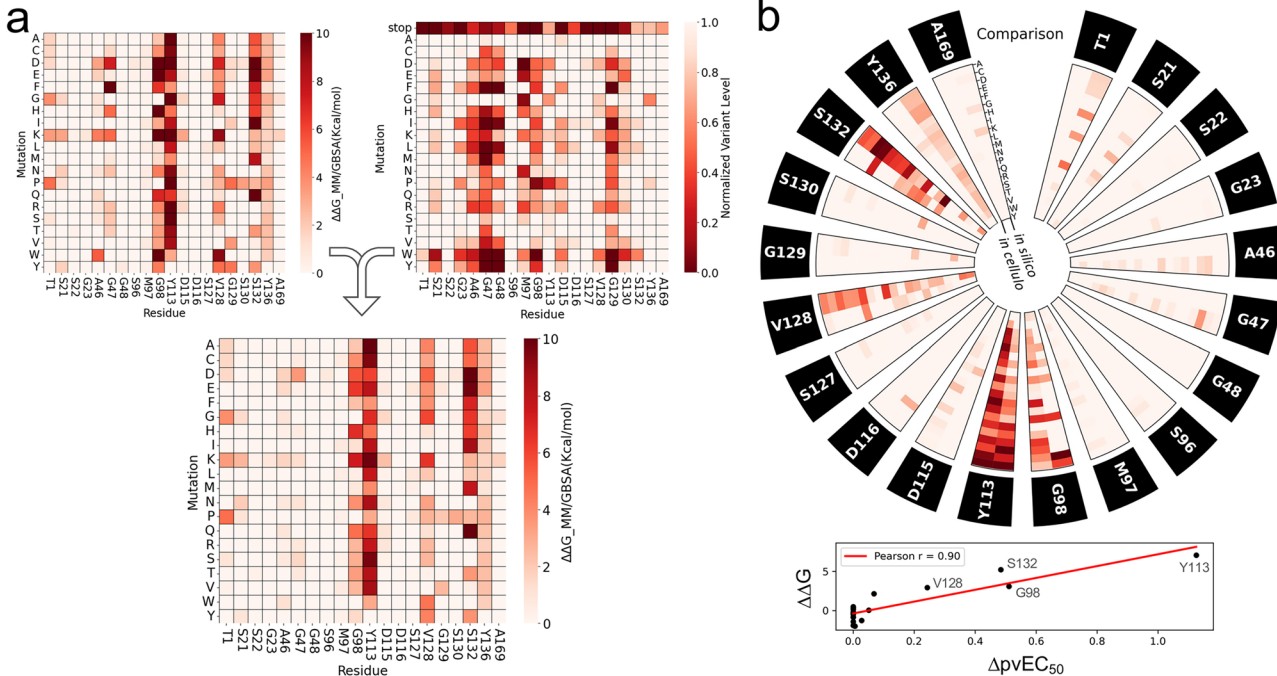

**Fig. 7 | Computational modelling combining affinity and function parameters.**
**a** The top left heatmap shows the impact of the full set of mutations on ligand affinity, as predicted using minimisation and MM/GBSA calculations, normalised to 0 as the baseline for the native sequence. The top right map shows the impact of the full set of mutations on proteasome function, as predicted using Evo 2. The lower heatmap shows predicted DDD247-resistance as determined by combining loss of ligand affinity predictions with penalties for loss of protein function predictions. **b** The circular plot compares the in cellulo measures of DDD247-resistance (virtual $EC_{50}$ values from Fig. 3a and Source Data, sheet 4) and the in silico prediction of DDD247-resistance from the lower heatmap in (**a**). The correlation plot shows the same data averaged for all mis-sense mutations at each site, with resistance hotspots identified.

sites, we can improve our understanding of mutations associated with drug resistance, improve prospects for resistance surveillance, and enhance computational drug design strategies towards more effective and durable drugs.

## Methods

### *T. brucei* growth and manipulation

Bloodstream form Lister 427 *Trypanosoma brucei* wild-type and *MSH2* RNAi[15] strains were cultured in HMI-11 (Gibco) supplemented with 10% fetal bovine serum (Sigma) at 37 °C and with 5% $CO_2$ in a humidified incubator. Genetic manipulations was carried out by electroporation using a Nucleofector (Lonza), with a Human T-cell kit (Lonza), or with custom-made transfection buffer[34], with the Nucleofector set to Z-001 (Amaxa).

### Oligo targeting

Oligo targeting was carried out using single-stranded oligodeoxynucleotides (ssODNs, Thermo Fisher Scientific) essentially as described[15] (Source Data, sheet 1). Briefly, for routine assays, we used 40 μg of each ssODN in 10 μl of 10 mM Tris-Cl, pH 8.5, mixed with 25 million cells in 100 μl transfection buffer. Selection with DDD247 or DDD248, compounds 7a and 7, respectively in Wyllie et al.[21], was applied 6 h later, as appropriate, and cells were subjected to limiting dilution in 96-well plates as required. Genomic DNA was extracted using a Qiagen DNeasy Kit followed by PCR amplification of the β5 gene (25 cycles). PCR products were purified using a PCR Purification Kit (Qiagen). To identify individual edits, amplicons were generated using Phusion polymerase (NEB) and amplicons were subjected to Sanger sequencing (Genewiz, Azenta Life Sciences).

### Assembly and screening of multiplexed oligo targeting libraries

For site saturation mutagenesis using a set of twenty degenerate ssODNs, we used 40 μg of each ssODNs in 10 μl of 10 mM Tris-Cl, pH

8.5, mixed with 6.5 million cells in 100 μl transfection buffer. *MSH2* knockdown was induced 24 h prior to transfection by the addition of 1 μg/mL tetracycline (Sigma). Each ssODN was transfected individually and the cultures were then pooled in 150 ml of medium to generate the library, comprising ~65 million cells based on an electroporation survival rate of ~50%. After 6 h, cells from 20 ml of the culture were cryo-preserved, cells from another 20 ml were collected for (pre-selection) DNA extraction, and the remainder were split into two cultures and subjected to selection with 8 nM compound DDD247. Following 4–5 d, 50 ml of each culture was collected for DNA extraction, and the remainder was split into three cultures that were subjected to selection with either 40 nM, 200 nM or 1 μM compound DDD247. Following 2–3 d, each culture was collected for DNA extraction.

For selection with bortezomib (Sigma), a cryo-preserved library sample was thawed, allowed to recover for 24 h, and split into two parallel cultures. Bortezomib selection was applied at 4 nM ($EC_{50} = 2.2$ nM). Following 7–9 d, cultures were collected for DNA extraction and the remainder was split into three cultures that were subjected to selection with higher concentrations of bortezomib; none of these latter cultures yielded resistant cells.

For fitness profiling, we first generated a *T. brucei MSH2* RNAi strain with a single β5 allele. A construct was synthesized comprising a *BLA* (blasticidin deaminase) selectable marker, flanked by *T. brucei* tubulin mRNA processing sequences and ~300 bp of *β5* gene flanking sequence to promote homologous recombination. This cassette was excised from the plasmid using AscI and used to transfect *MSH2* RNAi cells. *BLA* selection was applied at 10 μg/ml after 6 h and positive clones were assessed using PCR assays. Multiplex oligo targeting libraries were then assembled in both strains, either haploid or diploid for the *β5* gene, as described above, and again using the same set of twenty degenerate ssODNs. In this case, cells were collected for DNA extraction after 6 h and 4d without any drug selection. To identify multiplexed edits, PCR amplicons from the β5 gene were generated

using KOD Hot Start DNA Polymerase (Merck). We used 0.5 µg of genomic DNA template to assess library coverage and 0.1 µg of genomic DNA template to assess drug-selected samples; 25 cycles in all cases. Amplicons were purified using a PCR Purification Kit (Qiagen).

## Amplicon sequencing and codon variant scoring

Amplicons were subjected to deep sequencing using DNBseq (BGI Genomics); the controls and drug-selected samples shown in Figs. 1 and 2 were sequenced in parallel. Filtering of sequencing reads was performed using the SOAPnuke software with parameters: "-n 0.001 -l 10 -q 0.4 --adaMR 0.25 --ada_trim --minReadLen150" for drug selection experiments and "-n 0.01 -l 20 -q 0.3 --adaMR 0.25 --ada_trim --polyX 50 --minReadLen 150" for the haploid v diploid experiment.

Codon variant scoring was performed with the OligoSeeker (0.0.5) Python package[35] designed to process paired FASTQ files and count occurrences of specific codons. To visualise codon variant scores, we performed a normalization step by dividing each codon variant score by the total reads for that position and converting the fraction of reads to a percentage. This was followed by background correction whereby we subtracted the values for the control sample. Negative values were replaced with 0, and average values for the duplicate libraries were then calculated to give the codon variant scores. Codons registering elevated read-counts in the control sample, including all single nucleotide variants, were excluded from the analyses shown in Figs. 1c and 4c. To derive virtual $EC_{50}$ values, we derived a weighted average of codon variant scores across the DDD247 compound dose range. Briefly, the drug concentration associated with the highest read-count was assigned a weighting of '1' and relative weighting was assigned accordingly. For example, the virtual $EC_{50}$ for $G^{98}A$ is the average of 0.48*8, 1*40, 0.34*200, and 0.02*1000 = 32 nM. To visualize relative codon frequency using the pyCirclize python package (https://github.com/moshi4/pyCirclize, v1.9.1), raw codon counts were normalized by the maximum values and replicates were then averaged.

## Assembly of a panel of spontaneous drug-resistant mutants

We exploited the *MSH2* RNAi strain to rapidly generate a panel of spontaneous and independent DDD247-resistant mutants. To ensure that each mutant was independent, we first used limiting dilution to isolate twenty clones from the *MSH2* RNAi cell line. *MSH2* knockdown was induced in each clone for 24 h, tetracycline was removed by washing in HMI-11 medium, and compound DDD247 was added at 40 nM or 100 nM. We observed cell death followed by recovery and obtained twenty-three independent DDD247-resistant cultures using this approach. We sub-cloned each culture by limiting dilution, and isolated one clone from each for DNA extraction, PCR amplification of the β5 gene and Sanger sequencing (Genewiz, Azenta Life Sciences). We also subsequently PCR amplified and sequenced the proteasome β4 gene in three clones for which no mutations were identified in the β5 gene.

## Dose-response assays

To determine the Effective Concentration of drug to inhibit growth by 50% ($EC_{50}$), cells were plated in 96-well plates at $1 \times 10^3$ cells/ml in a 2-fold serial dilution of selective drug. Plates were incubated at 37 °C for 72 h, 20 µl resazurin sodium salt (AlamarBlue, Sigma) at 0.49 mM in PBS was then added to each well, and plates were incubated for a further 6 h. Fluorescence was determined using an Infinite 200 pro plate reader (Tecan) at an excitation wavelength of 540 nm and an emission wavelength of 590 nm. $EC_{50}$ values were derived using Prism (GraphPad).

## Homology modelling, docking and molecular dynamics

A *T. brucei* proteasome homology model was generated using a cryo-EM structure of the *Leishmania* proteasome in complex with the GSK3494245 inhibitor (PDB: 6qm7)[21], and the homology modelling tool implemented in Maestro (Schrodinger inc.)[36]. Docking was performed with Glide XP[37], using GSK3494245 for validation, followed by compound DDD247, with RMSD comparisons calculated using an MCS protocol. The complex was placed in a Cubic box with waters and ions (0.15 M) using Maestro and the system was subjected to 80 ns of molecular dynamics with Desmond[38]. The trajectory was clustered by RMSD (resulting in 9 clusters), and representative frames were selected for further analysis. Those clusters were classified with interaction fingerprints[39].

## Free energy estimations

The ensemble of conformations was generated from the clusters produced by the simulation to estimate the binding free energy. The residue scanning tool implemented in Maestro was used to compute approximate binding energy scores between the ligand and wild-type/mutant proteins[38]. This tool implements a high-throughput workflow for testing mutations with the ΔΔG associated (calculated with the MM/GBSA method)[29] using wild-type as reference (baseline). MM/GBSA is a free energy method that gives an approximation of relative binding free energy between states (such as ligand bound vs unbound). The method decomposes the calculation into terms; Van der Waals, Coulombic, hydrogen bonding, solvation and hydrophobic, for contributions to the final ΔG. All residues tested experimentally were mutated to build single mutants, with minimisation around 7 Å as the final step of preparation. Results were plotted with Seaborn and Matplotlib. For virtual $EC_{50}$ values derived from MOT-library profiling data, a ΔpvEC50 was calculated for each mutant, and plotted against the ΔΔG values.

## Protein function prediction

Machine learning approaches can effectively predict the impact of mutations on protein function[40] and we used an Evo 2 model to predict loss of proteasome function here (evo2_7b_base)[30]. Taking native nucleotide sequence for the *β5* subunit gene, we considered all possible mutations in those codons surveyed by MOT-library profiling. After the inference, the native sequence was used as a reference, and sequences with mutations were scored and subtracted from the reference score to obtain Evo2 Δ_scores. These scores were then used to apply penalties to the affinity predictions. Values were normalised to allow comparison of in cellulo and in silico predictions and the results were visualised in a circular plot with Seaborn and Matplotlib.

## Data availability

The high-throughput sequencing data generated for this study have been deposited at the Sequence Read Archive under accession code PRJNA1295514 (https://www.ncbi.nlm.nih.gov/bioproject/1295514). Source data for Figs. 1c, 2b–d, 3a, 4b, c and 5a, b are available as a Source Data file and for Figs. 6c, d, f, g, 7a, b, Supplementary Figs. 7c and 10a,b in the Github repository (https://github.com/velocirraptor23/Decoding-efficacy-and-resistance-space-at-a-drug-binding-site-/tree/main) and Zenodo https://zenodo.org/records/18195459. The homology model used in Figs. 6 and 7 is available in ModelArchive (https://modelarchive.org/doi/10.5452/ma-cxis8).

## Code availability

The code used to develop the homology model, perform analyses and generate results in this study is publicly available and has been deposited in Github at https://github.com/velocirraptor23/Decoding-efficacy-and-resistance-space-at-a-drug-binding-site-/tree/main, under MIT license. The specific version of the code associated with this publication is archived in Zenodo and is accessible via https://zenodo.org/records/18195459[41].

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

## Acknowledgements

We thank David Robinson for assistance with structural analysis and Anna Creelman and Hayley Bell for assistance with PCR and sequencing. This work was supported by a Wellcome Centre Award (223608/Z/21/Z to D.H. as co-applicant) and a Wellcome Investigator Award (217105/Z/19/Z to D.H.).

## Author contributions

The in cellulo experiments were designed by S.A., M.R. and D.H. and carried out by S.A. and M.R. Compound selection was carried out by M.D.R and M.T. Sequence data analysis was performed by M.T. Data analyses were performed by S.A., M.T. and D.H. Computational

modelling was performed by C.M-M, J.S.S. and P.E.G.F.I. The work was supervised by D.H. and M.J.B. The manuscript was written by S.A., C.M-M. and D.H. The manuscript was edited by all authors.

## Competing interests

The authors declare no competing interests.
