## [Transparent Peer Review file · Nature Communications]

Decoding efficacy and resistance space at a drug binding site

Corresponding Author: Professor David Horn

Version 0:

Reviewer comments:

Reviewer #1

(Remarks to the Author)

In this work, the authors present a systematic mapping of the proteasome $\beta 5$ subunit mutations conferring resistance to the inhibitor DDD247 using Multiplex Oligo Targeting (MOT). The key contributions lie in 1. the accurate design of the pooled MOT libraries with massive codon edits across residues in $\beta 5$ subunit binding pockets. 2. High-resolution experimental profiling of resistance, including virtual EC50 estimation directly from pooled deep mutational scans. 3. Identification of four major resistance hotspots (e.g., Y113, G98, V128, S132), with codon-level granularity of tolerated vs. non-tolerated substitutions. 4. Integration of Evo2 predictions (sequence-based functional tolerance) with docking-based ligand affinity modelling to propose a combined predictive framework for resistance.

From the computational side, the integration of Evo2, along with other computational approaches, is novel but the results are quite qualitative and are not significant and solid enough to support a unique novelty. Adding the following details would significantly strengthen the article:

1. Virtual EC50 derivation:

* Please provide more detail on how virtual EC50 values were computed from pooled sequencing data.

* How was the Hill slope (n) handled, given that only four drug concentrations were tested? Was it fixed at 1? How sensitive the correlation is to this hyperparameter.

2. The bespoke mutant validation set appears to be enriched for strong resistance alleles. This partly explains the near-perfect correlation. Please discuss this validation bias and, if possible, include weaker or borderline variants to test the robustness of virtual EC50 inference.

3. Evo2 evaluation: the contribution of Evo2 is argued qualitatively.

I encourage adding quantitative benchmarks comparing: a. Docking-only predictions, b. Evo2-only predictions, and c. Docking+Evo2 combined. Suggested metrics include correlation to (virtual) EC50, AUROC/AUPRC against a binary resistance definition, and hotspot overlap with experimental data.

4. Figure 7 lacks quantitative metrics. It is difficult to judge whether predictions and experiments align. For example, a scatter plot of average predicted vs experimental resistance per residue would be helpful. Figure 6h is on the right track. However, the analysis is limited to selected variants and only reports R2. I encourage the authors to extend this comparison to the full dataset and include additional metrics such as Spearman correlation, AUROC/AUPRC, and site-level hotspot overlap (i.e., Experimentally observed hotspots (Y113, G98, etc) vs predicted hotspots from docking+Evo2.)

Although the computational analyses could be strengthened, the genetic and experimental work of the paper is strong and novel.

(Remarks on code availability)

I have the following comments about the repository:

Please provide detailed descriptions for the MD_simulation, data, and MD_clusters subfolders, including:

- * Purpose and contents of each directory

The main README file should include:

- * A comprehensive introduction to each subfolder's role in the analysis pipeline
- * A step-by-step execution guide that walks through the entire workflow

Reviewer #2

(Remarks to the Author)

Altmann et al have conducted a comprehensive and impressive study employing oligo targeting to perform nearly complete saturation mutagenesis at the native drug-binding site of the *Trypanosoma brucei* proteasome beta-5 subunit, successfully generating detailed resistance and fitness landscapes. By combining high-throughput mutational screening, resistance profiling, and computational modelling, they demonstrate both the accuracy of their approach and its potential to predict spontaneous resistance mutations, providing valuable insights into the constraints shaping resistance evolution. The work significantly advances the methodology for mutagenesis in protozoan parasites and offers novel biological and pharmaceutical insights, especially regarding the identification of resistance hotspots and the interplay between fitness and resistance. Overall, this represents a major step forward for drug target characterization in neglected parasitic diseases.

Key findings include the successful application of multiplex oligo targeting to saturate 20 residues within the drug-binding pocket, the identification of over 100 resistance-conferring mutations with associated fitness effects, and the validation that these mutations mirror spontaneous resistance mutations observed in vitro and across species. Further, the integration of structural modelling and AI-driven predictions supports the empirical data, highlighting the potential for this approach, though the correlation between computed and experimental data remains moderate ($R^2 = 0.42$). But this limitation is acknowledged.

I have only minor comments:

1. Lines 50-70: The authors effectively discuss the limitations of CRISPR-based editing and the advantages of oligo targeting. However, it would be helpful to include additional information about the scalability of this approach. Specifically, how large a library can realistically be generated, and how many sites can be targeted simultaneously, given current system limitations? A brief comparison of this scalability to what is achievable with CRISPR-based or base-editing screens would also strengthen this section.
2. Figure 2a should include the untreated control condition for clarity and ease of interpretation.
3. It would also be useful to briefly restate the rationale for using the MSH2 RNAi cell line, even if this was covered in Altmann et al. (2022), to make the manuscript more self-contained.
4. A concise summary paragraph highlighting the key new findings and advances compared with previous literature would help readers clearly understand the novel contributions of this study. This is particularly important given the extensive existing literature on the *T. brucei* proteasomes.
5. Lines 510-517: While the authors provide clear evidence that fitness profiling is feasible using their approach, it would be helpful for others if more methodological detail were included on how this was achieved without additional selection and despite the relatively low transfection efficiencies previously reported for oligo targeting. For example, could the authors clarify what the effective representation of the transfected library was (i.e., the number of cells per construct) and how many total cells were transfected to achieve this level of representation? I am also uncertain whether the library used for the fitness screen was actually subjected to 8 nM DDD247 selection. In the Methods section, the authors describe a general library transfection followed by compound selection, yet in the main text (line 197) they state that fitness profiling was performed "without any drug selection". It appears that the fitness profiling did not involve drug treatment, but if so, sequencing depth must have been very high to achieve sufficient resolution of fitness profiles. Providing these details would substantially improve clarity.
6. Finally, the library design does not seem to include a "non-targeting" control, which is typically incorporated into CRISPR or similar screening libraries to demonstrate assay specificity. While the authors do validate specificity by profiling the library with bortezomib and present other supporting data that increase confidence in the results, it would be useful if they could briefly comment on why such a non-targeting control was not included or whether they would include such controls in the future.

(Remarks on code availability)

While I have not tested the code, the GitHub repository appears well organized and clearly documented. It should be

straightforward to reproduce the presented data and to follow the provided instructions for applying the workflow to similar datasets.

Reviewer #3

(Remarks to the Author)

Altmann et al describe the utilization of multiplex oligo targeting, deep sequencing and computational modeling to identify mutations that confer resistance and loss-of-function phenotype by carrying out saturation mutagenesis of selected interactions of the molecule (inhibitor) binding to target. They demonstrate the use of this workflow by utilizing known proteasome inhibitor and elegantly identify key mutations that are leading to robust resistance hotspots. This work integrates empirical resistance/fitness landscapes with in-silico docking, molecular dynamics clustering, and Evo2 functional predictions, demonstrating a combined framework to predict relative resistance and constrain mutational space by fitness.

Below are some of the potential points to consider and discuss

1. Generalizability of findings: The work is centered on a single compound (DDD247) and one parasite species *T. brucei*; resistance landscapes may differ across related inhibitors or species. It would have been interesting to show, if these mutations can be validated by another proteasome inhibitor with a different chemical structure. Although, there is high likelihood that these mutations do confer resistance to other structurally different inhibitor.
2. Co-editing artifacts: Authors have pointed out initial pooled ssODN approach can introduce multiple edits per cell, potentially confounding genotype–phenotype links. Although, the switch to individual delivery helps, but residual co-editing risk and selection bottlenecks may need to be carefully controlled.
3. Heterozygous versus homozygous: The technology used is very helpful for targets which depend heterozygous mutations are dominant thereby showing resistance. If the resistance needs homozygosity (mutations in both alleles), this method might not be that useful. Authors could highlight this limitation in their discussion.
Allele-specific effects: *T. brucei* is diploid and the current technology, typically edits a single allele; heterozygosity may mask fitness costs or resistance phenotypes compared to full homozygous states.
4. Selection biases: Stepwise drug selection can enrich fitter mutants while missing rare but relevant variants; I am hoping this issue might not be of big concern given the extensive efforts made by authors using varying drug concentration and deep sequencing in order to isolate all the possible mutations.
5. Modeling assumptions: Combining Evo2 functional scores with docking/free-energy predictions involves thresholds and normalization choices that may influence classification of loss-of-function vs resistance; Very heartening to see significant correlation between virtual EC50s and measured EC50s.
6. Structural inference: Homology modeling and docking rely on a related proteasome structure; uncertainties in local side-chain positioning could affect predicted interaction fingerprints.
7. The current methodology may not be able to capture structural changes that may affect compound binding, if the mutations far ($>5\text{\AA}$), thereby leading to compound resistance and loss-of-fitness.

Overall, the work appears methodologically solid and well-documented, with clear resistance hotspots identified. The main caveats are the scope (single target/compound/species) and potential biases from editing and selection workflows, which are common challenges in mutational scanning studies.

The MOT and computational modeling has potential not only to enable drug design but also to monitor resistance surveillance.

Minor comments

Line 23-24: the sentence ending “however”. May be you can start with “However,”

Line 171-172: Authors refer to “forty-six distinct drug-resistant mutants, which displayed up to 70- fold increased resistance”. I was unable to find this tables. Please refer to Table.

(Remarks on code availability)

Version 1:

Reviewer comments:

Reviewer #1

(Remarks to the Author)

The authors have addressed my main concerns regarding the computational validation and the methodology. The quantitative analysis is now sufficiently clear. Specifically:

- The clarification of the virtual EC_{50} calculation as a weighted average is reasonable and addresses my concern about curve fitting parameters².
- I reviewed the supplementary data provided in response to the selection bias concern. I am satisfied that the high correlation in Figure 3 is robust and not an artifact.
- I appreciate the authors' transparency regarding Evo2's low standalone predictive power ($r = 0.01$). Please ensure this specific limitation is explicitly stated in the main text so the model's contribution is not overstated.

(Remarks on code availability)

The documentation for the code is improved this time. However, for b) Free Energy MMGBSA/Calculations, the current instructions provided are minimal ('run externally using the Schrodinger suite'). It is recommended to include more detailed information.

Reviewer #2

(Remarks to the Author)

All my previous comments have been addressed. I recommend the manuscript for publication.

(Remarks on code availability)

Reviewer #3

(Remarks to the Author)

Authors have addressed all the comments raised by me. I am satisfied with the changes made.

(Remarks on code availability)

I am not expert in reading the codes. Hence, I am unable to comment on the code provided

Responses to Reviewer Comments

Reviewer #1:

In this work, the authors present a systematic mapping of the proteasome $\beta 5$ subunit mutations conferring resistance to the inhibitor DDD247 using Multiplex Oligo Targeting (MOT). The key contributions lie in 1. the accurate design of the pooled MOT libraries with massive codon edits across residues in $\beta 5$ subunit binding pockets. 2. High-resolution experimental profiling of resistance, including virtual EC₅₀ estimation directly from pooled deep mutational scans. 3. Identification of four major resistance hotspots (e.g., Y113, G98, V128, S132), with codon-level granularity of tolerated vs. non-tolerated substitutions. 4. Integration of Evo2 predictions (sequence-based functional tolerance) with docking-based ligand affinity modelling to propose a combined predictive framework for resistance.

From the computational side, the integration of Evo2, along with other computational approaches, is novel but the results are quite qualitative and are not significant and solid enough to support a unique novelty. Adding the following details would significantly strengthen the article:

1.1: Virtual EC₅₀ derivation:

* Please provide more detail on how virtual EC₅₀ values were computed from pooled sequencing data.

* How was the Hill slope (n) handled, given that only four drug concentrations were tested?

Was it fixed at 1? How sensitive the correlation is to this hyperparameter.

R1.1: We've added further detail to the Methods section, which now states "Briefly, the drug concentration associated with the highest read-count was assigned a weighting of '1' and relative weighting was assigned accordingly. For example, the virtual EC₅₀ for G⁹⁸A is the average of $0.48 \cdot 8$, $1 \cdot 40$, $0.34 \cdot 200$, and $0.02 \cdot 1000 = 32$ nM". We've also added "see Methods" at the relevant point in the Results section.

1.2: The bespoke mutant validation set appears to be enriched for strong resistance alleles. This partly explains the near-perfect correlation. Please discuss this validation bias and, if possible, include weaker or borderline variants to test the robustness of virtual EC₅₀ inference.

R1.2: The mean virtual EC₅₀ is 94 nM (n = 46) and the mean conventional EC₅₀ is 91 nM (n = 9); see Supplementary Data File (sheet 4). Thus, the mutant validation set is not significantly enriched for strong resistance alleles and there's no significant validation bias ($p = 0.94$).

1.3: Evo2 evaluation: the contribution of Evo2 is argued qualitatively.

I encourage adding quantitative benchmarks comparing: a. Docking-only predictions, b. Evo2-only predictions, and c. Docking+Evo2 combined. Suggested metrics include correlation to (virtual) EC₅₀, AUROC/AUPRC against a binary resistance definition, and hotspot overlap with experimental data.

R1.3: We thank the reviewer for this suggestion and have now used quantitative data for the contribution of Evo2 data to the predictions shown in Fig. 7, with further details included in Supplementary Fig. 10. We did not expect to see a strong correlation between Evo2 only predictions and our *in cellulo* resistance profile, partly because we believe that many "edits fail to yield drug resistance in our screens ... because they fail to impact drug binding affinity". Indeed Pearson's r is 0.01 in this case. We were reassured, however, to see that predicted loss-of-fitness was significantly lower for the set of 46 mutations we linked to drug resistance, relative to all 334 other possible mis-sense mutations ($p = 0.01$; Student's t -test). We've now added "Pearson's r for the full dataset was increased from 0.46 to 0.51 after applying penalties for predicted loss-of-fitness" to the Results text.

1.4: Figure 7 lacks quantitative metrics. It is difficult to judge whether predictions and experiments align. For example, a scatter plot of average predicted vs experimental resistance per residue would be helpful. Figure 6h is on the right track. However, the analysis is limited to selected variants and only reports R2. I encourage the authors to extend this comparison to the full dataset and include additional metrics such as Spearman correlation, AUROC/AUPRC, and site-level hotspot overlap (i.e., Experimentally observed hotspots (Y113, G98, etc) vs predicted hotspots from docking+Evo2.)

R1.4: We've now included a scatter plot for average predicted vs experimental resistance per residue, and added "a correlation plot shows how effectively our *in silico* analyses predicted the *in cellulo* derived resistance hotspots (Fig. 7b, Pearson's $r = 0.9$)" to the Results text. The Fig. 7 legend also now states "The correlation plot shows the same data averaged for all mis-sense mutations at each site, with resistance hotspots identified".

Although the computational analyses could be strengthened, the genetic and experimental work of the paper is strong and novel.

Remarks on code availability:

1.5: I have the following comments about the repository:

Please provide detailed descriptions for the MD_simulation, data, and MD_clusters subfolders, including:

- * Purpose and contents of each directory

The main README file should include:

- * A comprehensive introduction to each subfolder's role in the analysis pipeline

- * A step-by-step execution guide that walks through the entire workflow

R1.5: Updated as requested.

Reviewer #2:

Altmann et al have conducted a comprehensive and impressive study employing oligo targeting to perform nearly complete saturation mutagenesis at the native drug-binding site of the Trypanosoma brucei proteasome beta-5 subunit, successfully generating detailed resistance and fitness landscapes. By combining high-throughput mutational screening, resistance profiling, and computational modelling, they demonstrate both the accuracy of their approach and its potential to predict spontaneous resistance mutations, providing valuable insights into the constraints shaping resistance evolution. The work significantly advances the methodology for mutagenesis in protozoan parasites and offers novel biological and pharmaceutical insights, especially regarding the identification of resistance hotspots and the interplay between fitness and resistance. Overall, this represents a major step forward for drug target characterization in neglected parasitic diseases.

Key findings include the successful application of multiplex oligo targeting to saturate 20 residues within the drug-binding pocket, the identification of over 100 resistance-conferring mutations with associated fitness effects, and the validation that these mutations mirror spontaneous resistance mutations observed in vitro and across species. Further, the integration of structural modelling and AI-driven predictions supports the empirical data, highlighting the potential for this approach, though the correlation between computed and experimental data remains moderate ($R^2 = 0.42$). But this limitation is acknowledged.

I have only minor comments:

2.1: Lines 50-70: The authors effectively discuss the limitations of CRISPR-based editing and the advantages of oligo targeting. However, it would be helpful to include additional information about the scalability of this approach. Specifically, how large a library can realistically be generated, and how many sites can be targeted simultaneously, given current system

limitations? A brief comparison of this scalability to what is achievable with CRISPR-based or base-editing screens would also strengthen this section.

R2.1: We felt that it was more appropriate to address this point in the Discussion. Several recent studies report CRISPR-based editing of the *EGFR* gene, but only Kim *et al* (*Nature Biotechnology*, 2025) provide precise figures for library size, and we therefore use this report as a comparator. We now include the following paragraph in our Discussion:

“The current *T. brucei* libraries incorporated 1,280 alternative codons encoding 400 proteasome β 5 subunit variants. In this case, our goal was to introduce every possible alternative codon at each of twenty targeted sites. Based on an allele replacement frequency of approx. 0.04 % following transient suppression of mismatch repair, we estimated 26,000 edited cells among 65 million cells in a 150 ml culture volume, which equates to an average twenty-fold redundancy for each alternative codon. For comparison, a library generated recently using prime editors in human cells incorporated 2,476 single nucleotide edits encoding 1,726 epidermal growth factor receptor variants (Kim *et al.*, 2025). In terms of scaling, our results suggest that larger libraries could readily be generated and screened using the oligo targeting approach by simply using larger numbers of degenerate ssODNs. Substantial further scaling would present challenges in terms of assessing every alternative codon at each targeted site, but use of larger cell numbers and culture volumes would reduce bottlenecks and maintain library complexity”.

2.2: Figure 2a should include the untreated control condition for clarity and ease of interpretation.

R2.2: The untreated control library is described in Fig. 1. To clarify, we added “described above” to the Fig. 1 legend. We also now note in the Methods section that “the controls and drug-selected samples shown in Fig. 1-2 were sequenced in parallel”.

2.3: It would also be useful to briefly restate the rationale for using the MSH2 RNAi cell line, even if this was covered in Altmann *et al.* (2022), to make the manuscript more self-contained.

R2.3: We previously restated the rationale on lines 114-115, but have now added ‘*MSH2*’ to this text to clarify: “Mismatch-repair, previously reported to suppress editing efficiency approx. 50-fold¹⁴, was transiently knocked down using *MSH2* RNA interference for 24 h”.

2.4: A concise summary paragraph highlighting the key new findings and advances compared with previous literature would help readers clearly understand the novel contributions of this study. This is particularly important given the extensive existing literature on the *T. brucei* proteasomes.

R2.4: We thank the reviewer for this suggestion and have added the following text to the summary paragraph at the end of the Discussion:

“We developed multiplex oligo targeting in *T. brucei* and used the approach to saturation edit a native drug binding site in the proteasome β 5 subunit. We profiled both fitness and resistance space and identified resistance hotspots and >100 resistance-conferring mutants, which allowed us to accurately predict spontaneous resistance mutations in multiple trypanosomatids. Assessment of the empirical data using computational modelling provided unprecedented insights into anti-trypanosomal proteasome inhibitor structure-activity relationships”.

2.5: Lines 510-517: While the authors provide clear evidence that fitness profiling is feasible using their approach, it would be helpful for others if more methodological detail were included on how this was achieved without additional selection and despite the relatively low transfection efficiencies previously reported for oligo targeting. For example, could the authors clarify what the effective representation of the transfected library was (i.e., the number of cells per construct) and how many total cells were transfected to achieve this level of representation? I am also uncertain whether the library used for the fitness screen was actually subjected to 8 nM DDD247 selection. In the Methods section, the authors describe a general library transfection followed by compound selection, yet in the main text (line 197) they state

that fitness profiling was performed “without any drug selection”. It appears that the fitness profiling did not involve drug treatment, but if so, sequencing depth must have been very high to achieve sufficient resolution of fitness profiles. Providing these details would substantially improve clarity.

R2.5: We've now added further detail in the Methods section; “the library, comprising approx. 65 million cells based on an electroporation survival rate of approx. 50 %”, and also more detail in the Discussion, including “we estimated an average twenty-fold redundancy for each alternative codon” (see R2.1 above). For the fitness profiling section, we added more detail to clarify in the Methods section; “In this case, cells were collected for DNA extraction after 6 h and 4 d without any drug selection”. We've also added detail in the Figure legends for sequencing depth, which was indeed higher for the controls shown in Fig. 1c (“average of >33 M reads mapped per site”) and for the fitness profiling shown in Fig. 4c (“average of >27 M reads mapped per site”), than for resistance profiling shown in Fig. 2d (“average of >6.5 M reads mapped per site”).

2.6: Finally, the library design does not seem to include a “non-targeting” control, which is typically incorporated into CRISPR or similar screening libraries to demonstrate assay specificity. While the authors do validate specificity by profiling the library with bortezomib and present other supporting data that increase confidence in the results, it would be useful if they could briefly comment on why such a non-targeting control was not included or whether they would include such controls in the future.

R2.6: Non-targeting control sgRNAs are typically included in CRISPR experiments to establish a baseline for off-target editing by Cas9. We feel that a non-targeting control oligo would be of limited value for oligo targeting, since a ‘zero’ baseline is already set by several targeting oligos, those that fail to yield any drug resistance associated edits.

Remarks on code availability:

While I have not tested the code, the GitHub repository appears well organized and clearly documented. It should be straightforward to reproduce the presented data and to follow the provided instructions for applying the workflow to similar datasets.

Reviewer #3:

Altmann et al describe the utilization of multiplex oligo targeting, deep sequencing and computational modeling to identify mutations that confer resistance and loss-of-function phenotype by carrying out saturation mutagenesis of selected interactions of the molecule (inhibitor) binding to target. They demonstrate the use of this workflow by utilizing known proteasome inhibitor and elegantly identify key mutations that are leading to robust resistance hotspots. This work integrates empirical resistance/fitness landscapes with in-silico docking, molecular dynamics clustering, and Evo2 functional predictions, demonstrating a combined framework to predict relative resistance and constrain mutational space by fitness.

Below are some of the potential points to consider and discuss

3.1: Generalizability of findings: The work is centered on a single compound (DDD247) and one parasite species *T. brucei*; resistance landscapes may differ across related inhibitors or species. It would have been interesting to show, if these mutations can be validated by another proteasome inhibitor with a different chemical structure. Although, there is high likelihood that these mutations do confer resistance to other structurally different inhibitor.

R3.1: We had previously noted that our data can predict spontaneous drug resistance mutations observed in other trypanosomatids (see lines 257-262) and have now added more detail to the current text regarding other inhibitors; “Y¹¹³F in *T. brucei*, which confers resistance to LXE408²³, D¹¹⁵N in *T. cruzi*, which confers cross-resistance to DDD248, GNF6702, and TCMDC-143194²⁵, and G⁹⁸C and G⁹⁸S in *Leishmania*, the latter of which confers resistance to DDD248⁷”; also now noting that our data can predict “spontaneous drug resistance

mutations in multiple trypanosomatids, and for multiple anti-trypanosomal proteasome inhibitors”.

3.2: Co-editing artifacts: Authors have pointed out initial pooled ssODN approach can introduce multiple edits per cell, potentially confounding genotype–phenotype links. Although, the switch to individual delivery helps, but residual co-editing risk and selection bottlenecks may need to be carefully controlled.

R3.2: A co-editing artefact would be expected to produce a background signal across tolerated codons, including synonymous codons. We see no evidence for this in Fig. 2c-d. We have now added more detail on library complexity and mitigation against bottlenecks, in both the Methods and Discussion sections (see R2.1 and R2.5 above).

3.3: Heterozygous versus homozygous: The technology used is very helpful for targets which depend heterozygous mutations are dominant thereby showing resistance. If the resistance needs homozygosity (mutations in both alleles), this method might not be that useful. Authors could highlight this limitation in their discussion.

Allele-specific effects: *T. brucei* is diploid and the current technology, typically edits a single allele; heterozygosity may mask fitness costs or resistance phenotypes compared to full homozygous states.

R3.3: We’ve added the following text to the Discussion: “We do not observe homozygous edits using oligo targeting here, and this may be a limitation for investigating some drug resistance mechanisms”.

Heterozygosity does not appear to mask resistance phenotypes in our study, since “heterozygous edits were sufficient to yield resistance in every case” in the bespoke set of nine pairs of mutants. Fitness costs, on the other hand, are ‘masked’ by heterozygosity (see Fig. 4b-c) in the sense that defective alleles are tolerated in a diploid background, but such defective alleles would not be expected to yield resistance since “mutational resistance space will be constrained by mutational fitness space”.

3.4: Selection biases: Stepwise drug selection can enrich fitter mutants while missing rare but relevant variants; I am hoping this issue might not be of big concern given the extensive efforts made by authors using varying drug concentration and deep sequencing in order to isolate all the possible mutations.

R3.4: Mutations associated with loss of proteasome function and substantial loss-of-fitness under drug pressure will inevitably have been lost from the population during screening. On the other hand, we’re confident that our approach does not miss relevant variants since “codons for the same amino acid edit yielded highly consistent results”.

3.5: Modeling assumptions: Combining Evo2 functional scores with docking/free-energy predictions involves thresholds and normalization choices that may influence classification of loss-of-function vs resistance; Very heartening to see significant correlation between virtual EC50s and measured EC50s.

R3.5: We agree and have now used quantitative data from Evo2 predictions and added more detail on this analysis (see R1.3-1.4).

3.6: Structural inference: Homology modeling and docking rely on a related proteasome structure; uncertainties in local side-chain positioning could affect predicted interaction fingerprints.

R3.6: We do comment on ‘limitations’ associated with computational modelling in the Discussion but also find it reassuring that our models “yield effective *in silico* predictions of drug target affinity and relative resistance”, meaning that *in silico* analysis can be fruitful even when structural data are limited.

3.7: The current methodology may not be able to capture structural changes that may affect compound binding, if the mutations far (>5Å), thereby leading to compound resistance and

loss-of-fitness.

R3.7: The current MOT libraries do indeed incorporate only mutations affecting residues <5Å from the drug binding site. Although results may differ depending upon the target, our analysis of spontaneous mutations in the proteasome β5 subunit indicated that “all [spontaneous] β5 subunit mutations were predicted by MOT-library profiling; and no β5 subunit mutations were identified in residues more than 5 Å from the drug binding site”. In terms of loss-of-fitness, and as noted above, mutations associated with loss of proteasome function and substantial loss of fitness under drug pressure will inevitably have been lost from the population during screening (see R3.4).

3.8: Overall, the work appears methodologically solid and well-documented, with clear resistance hotspots identified. The main caveats are the scope (single target/compound/species) and potential biases from editing and selection workflows, which are common challenges in mutational scanning studies. The MOT and computational modeling has potential not only to enable drug design but also to monitor resistance surveillance.

R3.8: We thank the reviewer for these comments and certainly intend to expand the scope in the future.

Minor comments

3.9: Line 23-24: the sentence ending “however”. May be you can start with “However,”

R3.9: We considered both alternatives, but prefer the original version.

3.10: Line 171-172: Authors refer to “forty-six distinct drug-resistant mutants, which displayed up to 70- fold increased resistance”. I was unable to find this tables. Please refer to Table.

R3.10: We now refer to “(Supplementary Data sheet 4)”.

Responses to reviewers:

Reviewer 1:

The authors have addressed my main concerns regarding the computational validation and the methodology. The quantitative analysis is now sufficiently clear. Specifically:

- The clarification of the virtual EC_{50} calculation as a weighted average is reasonable and addresses my concern about curve fitting parameters².
- I reviewed the supplementary data provided in response to the selection bias concern. I am satisfied that the high correlation in Figure 3 is robust and not an artifact.
- I appreciate the authors' transparency regarding Evo2's low standalone predictive power ($r = 0.01$). Please ensure this specific limitation is explicitly stated in the main text so the model's contribution is not overstated.

We've added "Pearson's r for loss-of-fitness alone was 0.01" to the last section in the Results text.

Remarks on code availability:

The documentation for the code is improved this time. However, for b) Free Energy MMGBSA/Calculations, the current instructions provided are minimal ('run externally using the Schrodinger suite'). It is recommended to include more detailed information.

We've now added the following detail to the repository: "Free energy MMGBSA/Calculations were performed for each of the clusters with mutations. For these calculations, we provide PDB files from each cluster; however, the calculation needs to be run externally using the Schrodinger suite (Residue Scanning Tool, 2023-2). We provide the final csv file with results ('all_combined_final.csv'). This tool implements a high-throughput workflow for testing mutations with the free energy associated (calculated with the MM/GBSA method) using wild-type as reference (baseline). For more details, please, refer to the paper associated with this repository."

Reviewer 2:

All my previous comments have been addressed. I recommend the manuscript for publication.

Reviewer 3:

Authors have addressed all the comments raised by me. I am satisfied with the changes made.

Remarks on code availability:

I am not expert in reading the codes. Hence, I am unable to comment on the code provided